# Comparative mRNA booster effectiveness against death or hospitalization with COVID-19 pneumonia across at-risk US Veteran populations

J. Daniel Kelly [1,2,3,4] ✉, Samuel Leonard[1], W. John Boscardin [3], Katherine J. Hoggatt[1,2], Emily N. Lum[1], Charles C. Austin[5,6], Amy Byers[1], Phyllis C. Tien[1,2], Peter C. Austin[7], Dawn M. Bravata[5,8,9] & Salomeh Keyhani[1]

Studies of comparative mRNA booster effectiveness among high-risk populations can inform mRNA booster-specific guidelines. The study emulated a target trial of COVID-19 vaccinated U.S. Veterans who received three doses of either mRNA-1273 or BNT162b2 vaccines. Participants were followed for up to 32 weeks between July 1, 2021 to May 30, 2022. Non-overlapping populations were average and high risk; high-risk sub-groups were age ≥65 years, high-risk co-morbid conditions, and immunocompromising conditions. Of 1,703,189 participants, 10.9 per 10,000 persons died or were hospitalized with COVID-19 pneumonia over 32 weeks (95% CI: 10.2, 11.8). Although relative risks of death or hospitalization with COVID-19 pneumonia were similar across at-risk groups, absolute risk varied when comparing three doses of BNT162b2 with mRNA-1273 (BNT162b2 minus mRNA-1273) between average-risk and high-risk populations, confirmed by the presence of additive interaction. The risk difference of death or hospitalization with COVID-19 pneumonia for high-risk populations was 2.2 (0.9, 3.6). Effects were not modified by predominant viral variant. In this work, the risk of death or hospitalization with COVID-19 pneumonia over 32 weeks was lower among high-risk populations who received three doses of mRNA-1273 vaccine instead of BNT162b2 vaccine; no difference was found among the average-risk population and age >65 sub-group.

Although there is broad consensus that boosters are effective against hospitalization and severe COVID-19 illness[1], clinicians and the public face unanswered questions as to the comparative effectiveness of mRNA boosters by product type, over time, and within high-risk populations[2–4]. Left unanswered, these issues have the potential to undermine public confidence and feed narratives reinforcing vaccine hesitancy. As of October 2022, comparative effectiveness studies of primary vaccination with mRNA-1273 or BNT162b2 demonstrated a

[1]San Francisco VA Medical Center, San Francisco, CA, USA. [2]Department of Medicine, University of California, San Francisco (UCSF), San Francisco, CA, USA. [3]Department of Epidemiology and Biostatistics, UCSF, San Francisco, CA, USA. [4]F.I. Proctor Foundation, UCSF, San Francisco, CA, USA. [5]Department of Veterans Affairs (VA) Health Services and Development (HSR&D) Center for Health Information and Communication (CHIC) and the Department of Medicine, Richard L. Roudebush VA Medical Center, Indianapolis, IN, USA. [6]Veterans Affairs Medical Center, Indianapolis, IN, USA. [7]Institute for Clinical Evaluative Sciences, Toronto, Ontario, Canada. [8]Department of Medicine, Indiana University School of Medicine, Indianapolis, IN, USA. [9]Regenstrief Institute, Indianapolis, IN, USA. ✉e-mail: dan.kelly@ucsf.edu

small absolute benefit favoring the mRNA-1273 vaccine, which increased over a 24-week period without detailed sub-group analyses of high-risk populations such as those with immunocompromising conditions[2–4]. As a result, public health messaging has focused on the substantial protection against COVID-19 hospitalization, regardless of vaccine product and membership in an at-risk population[4].

Vaccination and boosting have shifted the spectrum of clinical disease toward mild COVID-19 illness[5], even as protective immunity wanes[6,7]. As a result, many patients who are hospitalized for reasons other than COVID-19 (e.g., fracture, alcohol withdrawal, asymptomatic but screened positive) but have a positive test for COVID-19 are classified as having a COVID-related illness, a commonly studied outcome in the COVID-19 literature[8]. In contrast, studies with highly specific outcomes such as COVID-19 pneumonia are lacking in the literature but can provide high-quality evidence to demonstrate ongoing effectiveness of vaccines. Extending our understanding of booster effectiveness against highly specific outcomes will guide the decision-making of individuals who live in an era of abundant vaccines and concerns as to whether a booster product will change their risk.

In this work, a national cohort of Veterans was used in a target trial emulation design to compare mRNA booster effectiveness at preventing breakthrough COVID-19 and death or hospitalization due to COVID-19 pneumonia following vaccination and booster with three doses of BNT162b2 or mRNA-1273 vaccine.

## Results

From an initial group of 6,286,624 participants, the analytic cohort consisted of 1,703,189 participants who received an initial COVID-19 vaccination series, followed by a third dose (Supplementary Fig. 1). Among these participants, 1,205,701 (70.8%) were age ≥65 years, and 1,566,828 (92.0%) were male; 1,188,470 (69.8%) had high-risk co-morbid conditions, and 189,057 (11.1%) had immunocompromising conditions (Table 1, Supplementary Table 1). The largest amount of missingness (6.4%) occurred with the race variable.

Of 1,703,189 participants, 917,954 (53.9%) received three doses of mRNA-1273 vaccine and 785,235 (46.1%) received three doses of BNT162b2 vaccine. Between 1 July and 30 November 2021 (Delta predominant variant period), 1,099,808 were boosted (64.6%); the remaining 603,381 participants (35.4%) were boosted after 1 December 2021 (Omicron predominant variant period). Among high-risk populations, 13.5% were age ≥65 years, 74.6% had high-risk co-morbid conditions, and 11.8% had immunocompromised conditions.

### Main outcomes

Over a 32-week follow-up period, 22,848 of 1,703,189 developed breakthrough COVID-19 (149.3 events per 10,000 persons; 95% CI: 147.3, 152.1), and 1,649 died or were hospitalized with COVID-19 pneumonia (10.9 events per 10,000 persons; 95% CI: 10.2, 11.8) (Table 2, Supplementary Table 2).

Comparing participants in the overall cohort who received three doses of BNT162b2 with three doses of mRNA-1273 (BNT162b2 minus mRNA-1273), risk differences were estimated as 23.9 events of breakthrough COVID-19 per 10,000 persons (95% CI: 19.8, 29.9) and as 2.1 events of death or hospitalization with COVID-19 pneumonia per 10,000 persons (95% CI: 0.6, 3.3). These risk differences translated into large numbers needed to vaccinate: 415.5 for breakthrough COVID-19 (95% CI: 334.7, 505.3), and 4,936.6 for death or hospitalization with COVID-19 pneumonia (95% CI: 3,028.8, 16,670.3) (Table 2). Comparative effects of the third mRNA vaccine dose (BNT162b2 versus mRNA-1273) were sustained over eras of Delta and Omicron predominant variants (Supplementary Table 3).

### Average-risk sub-group

Among average-risk participants who received three doses of BNT162b2 compared with those who received three doses of mRNA-1273

(BNT162b2 minus mRNA-1273), there was no difference in absolute risk of breakthrough COVID-19 or of death or hospitalization with COVID-19 pneumonia (Table 2). When considering relative risks, however, the cumulative incidence of breakthrough COVID-19 was higher among those who received three doses of BNT162b2 (Cumulative incidence ratio [CIR]: 1.1; 95% CI: 1.0, 1.3) (Table 4). There was no relationship between cumulative incidence of hospitalization and booster product.

### High-risk sub-groups (non-overlapping)

Among all high-risk participants (age ≥65 years, high-risk co-morbid conditions, immunocompromising conditions), participants who received three doses of BNT162b2 were at higher cumulative incidence and absolute risk of breakthrough COVID-19 (CIR: 1.2; 95% CI: 1.2, 1.3; RD: 24.5; 95% CI: 18.7, 29.4) and of death or hospitalization with pneumonia (CIR: 1.3, 95% CI: 1.2, 1.4; RD: 2.2; 95% CI: 0.9, 3.6) than those who received three doses of mRNA-1273 (Tables 2 and 4). Comparing the high-risk participants with average-risk participants, additive interaction was present for death or hospitalization with COVID-19 pneumonia (RERI: 1.6) (Table 2).

Within high-risk populations, the magnitudes of the cumulative incidence ratio of death or hospitalization with COVID-19 pneumonia were similar across sub-groups while the magnitudes of the risk differences varied with an increased absolute risk of death or hospitalization with COVID-19 pneumonia in the sub-group of high-risk co-morbid conditions (RD: 2.6; 95% CI: 0.5, 4.4), and immunocompromising conditions (RD: 6.6; 95% CI: −1.2, 13.5) compared with the sub-group of age ≥65 years (RD: −0.3; 95% CI: −3.2, 2.1), (Table 3, Table 4).

Comparing the sub-groups of high-risk co-morbid or immunocompromising conditions with lower risk sub-groups, additive interaction was present for death or hospitalization with COVID-19 pneumonia (RERI: 1.4), but multiplicative interaction was not present (p = 0.50) (Table 3). Further interaction analyses focusing on the subgroup of immunocompromising conditions against other sub-groups showed that the presence of additive interaction (RERI: 1.1).

To prevent one case of hospitalization with COVID-19 pneumonia or death, the estimated number needed-to-vaccinate (NNV) with three doses of mRNA-1273 compared with three doses of BNT162b2 during a 32-week follow-up period would be 1,411.7 for those with immunocompromising conditions. This NNV of 1,411.7 was 2.7 times lower than the NNV of 3,869.6 for the high-risk co-morbid condition sub-group (Table 3).

### Cumulative incidence curves

In the overall cohort, the cumulative incidence of death or hospitalization due to COVID-19 pneumonia by booster product was similar until week 12 at which time the cumulative incidence curves diverged after 12 weeks as shown in Fig. 1. Compared with the overall cohort, these differences were smaller among those with age ≥65 years (Fig. 2), similar among those with high-risk co-morbid conditions (not immunocompromised) (Fig. 3), and greater among those with immunocompromising conditions (Fig. 4).

## Discussion

In this national US cohort of adults receiving care at VHA facilities, 32-week relative and absolute risk of hospitalization or death due to COVID-19 pneumonia was statistically higher among those who received three doses of BNT162b2 vaccine than those who received three doses of mRNA-1273 vaccine. The cumulative incidence diverged 12 weeks after the third dose. In analysis by sub-group however, there was a divergence in the pattern of findings. Although relative risks remained consistent, differences in absolute risk were increased in the high-risk population and driven by sub-group's risk profile (high-risk co-morbid or immunocompromising conditions >> average-risk or age ≥65 years), suggesting that the specific populations of patients with

**Table 1 | Characteristics of the boosted cohort by vaccination series[a]**

| | Overall | mRNA-1273 ×3 | BNT162b2 ×3 |
|---|---|---|---|
| N | 1,703,189 | 917,954 | 785,235 |
| Male, n (%) | 1,566,828 (92.0) | 852,107 (92.8) | 714,721 (91.0) |
| Female, n (%) | 136,361 (8.0) | 65,847 (7.2) | 70,514 (9.0) |
| Age, median (q1.0, q3) | 72 (63.0, 76.0) | 72 (64.0, 77.0) | 71 (61.0, 76.0) |
| 18.0–34, n (%) | 25,570 (1.5) | 10,609 (1.2) | 14,961 (1.9) |
| 35.0–49, n (%) | 115,597 (6.8) | 51,286 (5.6) | 64,311 (8.2) |
| 50.0–64, n (%) | 356,321 (20.9) | 179,032 (19.5) | 177,289 (22.6) |
| 65.0–74, n (%) | 625,953 (36.8) | 341,048 (37.2) | 284,905 (36.3) |
| >75.0, n (%) | 579,748 (34.0) | 335,979 (36.6) | 243,769 (31.0) |
| Race[b], n (%) | | | |
| American Indian or Alaska Native | 11,220 (0.7) | 5995 (0.7) | 5225 (0.7) |
| Asian | 24,503 (1.4) | 11,757 (1.3) | 12,746 (1.6) |
| Black or African American | 330,079 (19.4) | 148,949 (16.2) | 181,130 (23.1) |
| Missing | 108,998 (6.4) | 57,135 (6.2) | 51,863 (6.6) |
| More than one race | 13,469 (0.8) | 6939 (0.8) | 6,530 (0.8) |
| Native Hawaiian or other Pacific Islander | 15,155 (0.9) | 7812 (0.9) | 7343 (0.9) |
| White | 1,199,765 (70.4) | 679,367 (74.0) | 520,398 (66.3) |
| Hispanic or Latino Ethnicity (regardless of race), n (%) | 120,447 (7.1) | 66,661 (7.3) | 53,786 (6.8) |
| Married, n (%) | 1,018,655 (59.8) | 560,871 (61.1) | 457,784 (58.3) |
| Urban[c], n (%) | 1,170,612 (68.7) | 571,598 (62.3) | 599,014 (76.3) |
| BMI, median (q1.0, q3) | 29 (26.0, 33.3) | 29 (26.0, 33.3) | 29 (26.1, 33.4) |
| BMI_cat | | | |
| <18.5, n (%) | 11,710 (0.7) | 6275 (0.7) | 5435 (0.7) |
| 18.5–24.9, n (%) | 280,883 (16.5) | 151,927 (16.6) | 128,956 (16.4) |
| 25.0–29.9, n (%) | 589,295 (34.6) | 316,639 (34.5) | 272,656 (34.7) |
| ≥30.0, n (%) | 721,252 (42.3) | 385,192 (42.0) | 336,060 (42.8) |
| Unknown, n (%) | 100,049 (5.9) | 57,921 (6.3) | 42,128 (5.4) |
| Hypertension | 1,043,181 (61.2) | 578,501 (63.0) | 464,680 (59.2) |
| Diabetes | 554,540 (32.6) | 308,289 (33.6) | 246,251 (31.4) |
| CKD[d] | 355,021 (20.8) | 199,994 (21.8) | 155,027 (19.7) |
| IHD | 291,422 (17.1) | 166,077 (18.1) | 125,345 (16.0) |
| COPD bronchiectasis | 196,384 (11.5) | 114,691 (12.5) | 81,693 (10.4) |
| CHF | 96,188 (5.6) | 53,640 (5.8) | 42,548 (5.4) |
| Immunocompromised[e] | 120,111 (7.1) | 64,316 (7.0) | 55,795 (7.1) |
| Cancer–solid organ | 59,172 (3.5) | 31,527 (3.4) | 27,645 (3.5) |
| Severe CKD[f] | 43,843 (2.6) | 26,242 (2.9) | 17,601 (2.2) |
| Stroke TIA | 45,239 (2.7) | 24,295 (2.6) | 20,944 (2.7) |
| Dementia | 32,089 (1.9) | 18,086 (2.0) | 14,003 (1.8) |
| Cirrhosis | 26,480 (1.6) | 13,133 (1.4) | 13,347 (1.7) |
| Cancer[g]–lymphoma leukemia | 21,103 (1.2) | 11,804 (1.3) | 9299 (1.2) |
| Dialysis | 12,780 (0.8) | 6924 (0.8) | 5856 (0.7) |
| Cancer[g]–other | 10,752 (0.6) | 5425 (0.6) | 5327 (0.7) |
| Date of first dose of vaccination series, by 3.0-month period | | | |
| 12/2020–02/2021 | 1,117,531 (65.6) | 624,545 (68.0) | 492,986 (62.8) |
| 03/2021–05/2021 | 560,429 (32.9) | 282,717 (30.8) | 277,712 (35.4) |
| 06/2021–08/2021 | 22,708 (1.3) | 9782 (1.1) | 12,926 (1.6) |
| 09/2021–11/2021 | 2513 (0.1) | 907 (0.1) | 1606 (0.2) |
| 12/2021–02/2022 | 8 (0.0) | 3 (0.0) | 5 (0.0) |
| Current smoker, n (%) | 352,343 (20.7) | 190,127 (20.7) | 162,216 (20.7) |
| Alcohol use disorder[h], n (%) | 118,757 (7.0) | 61,581 (6.7) | 57,176 (7.3) |
| Substance use disorder[i], n (%) | 74,262 (4.4) | 37,537 (4.1) | 36,725 (4.7) |
| Housing problems[j], n (%) | 60,951 (3.6) | 30,528 (3.3) | 30,423 (3.9) |
| Median follow-up time (IQR), days | 188 (163.0, 213.0) | 180 (159.0, 202.0) | 199 (168.0, 224.0) |
| Median time elapsed between initial dose and booster (IQR), days | 270 (246.0, 295.0) | 280 (258.0, 301.0) | 258 (237.0, 282.0) |
| High-risk populations, n (%) | 1,592,704 (93.5) | 867,665 (94.5) | 725,039 (92.3) |

**Table 1 (continued) | Characteristics of the boosted cohort by vaccination series[a]**

|  | Overall | mRNA-1273 ×3 | BNT162b2 ×3 |
|---|---|---|---|
| Age >65.0 and no high-risk co-morbid conditions, n (%) | 215,177 (12.6) | 118,313 (12.9) | 96,864 (12.3) |
| High-risk co-morbid conditions (not immunocompromised), n (%) | 1,188,470 (69.8) | 647,603 (70.5) | 540,867 (68.9) |
| Immunocompromised[e], n (%) | 189,057 (11.1) | 101,749 (11.1) | 87,308 (11.1) |
| Age <65.0 and no co-morbid conditions, n (%) | 110,485 (6.5) | 50,289 (5.5) | 60,196 (7.7) |

[a]Full Table 1 data found in Supplement Supplementary Table 2.
[b]Race/ethnicity was assessed using self-identified data found in Veteran Health Records.
[c]Urban/rural was assessed using defined based on the Rural Urban Commuting Area (RUCA) categories developed by the Department of Agriculture and Health and Human Services' Health Resource and Services Administration.
[d]CKD defined as having a glomerular filtration rate between 30 and 60.
[e]Immunocompromised definition based on medications and history of cancer (see supplement for list of meds).
[f]Severe CKD defined as having a glomerular filtration rate <30.
[g]Cancer definition based on diagnosis codes. 2 outpatient or 1 inpatient diagnosis code in the VHA (see supplement).
[h]Alcohol use disorder defined as 1 outpatient or 1 inpatient code within 2 years of index.
[i]Including cannabis, opioids, inhalants.
[j]Housing problems defined as homelessness, inadequate housing, other problems related to housing and economic circumstances.
*BMI* body mass index, *CHF* chronic heart failure, *CKD* chronic kidney disease, *COPD* chronic obstructive pulmonary disease, *TIA* transient ischemic attack.

high-risk co-morbid or immunocompromising conditions will gain from receiving three doses of mRNA-1273 vaccine instead of three doses of BNT162b2 vaccine.

Studies from this and other research groups have highlighted how those with immunocompromising conditions are at higher-risk of COVID-19 hospitalization than other at-risk groups such as those with high-risk co-morbid conditions[8,9], but have not compared the effect of booster products on COVID-19 hospitalization by immunocompetent and immunocompromised populations[6]. Further, studies comparing two doses of mRNA-1273 and BNT162b2 vaccine demonstrated similar magnitudes of relative risk across age groups and comorbidities as well as small absolute risk differences in the overall cohorts[2,3], but also did not include a group with immunocompromising conditions, which would have allowed for a direct comparison of absolute risk and NNV against other sub-groups. In our study, the immunocompromised sub-group had the largest magnitude of the risk difference of death or hospitalization with COVID-19 pneumonia; although non-significant and under-powered, this important finding meant that the NNV of 1411.7 among the immunocompromising conditions group was 2.7 times lower than the NNV of 3,869.6 among the high-risk co-morbid conditions group, suggesting that there may be a larger public health benefit among the immunocompromised group.

Even though the NNV can be used to inform tailored public health decisions, many other studies have focused on relative risks because of their value in estimating vaccine efficacy or effectiveness, ease of interpretation by the public[10], higher concordance across baseline risks[11], and the large numbers of NNVs associated with COVID-19 vaccines. One of the first major COVID-19 vaccine surveillance studies to be published reported a risk difference of 0.22 per 1,000 persons, which translated to a NNV of 4545 to prevent one COVID-19 hospitalization among unvaccinated individuals, were these persons to have been vaccinated[12]. Compared with an estimated vaccine effectiveness of 87% against COVID-19 hospitalization, such a high NNV may have been less appealing to clinicians and the public; however, the presentation of multiple effect estimates, inclusive of NNVs, can answer critical questions concerning a particular vaccine's differing public health benefits across various at-risk sub-groups[13]. In a later comparative effectiveness study of two doses each of mRNA-1273 vaccine and BNT162b2 vaccine, Dickerman, et al. reported the estimated NNV with mRNA-1273 instead of BNT162b2 would be 1818 (95% CI: 1,205.0, 2,778.0) to prevent one case of COVID-19 hospitalization[2]. The NNVs in our study were roughly similar to other studies comparing mRNA-1273 and BNT162b2 vaccines, except that our study examined a more specific outcome (death or hospitalization with COVID-19 pneumonia). Further, the high-risk population had relatively lower NNVs than the average-risk population. The non-significant absolute risk differences of COVID-19 hospitalization among average-risk population and the age >65 sub-group in our study should provide reassurance as to the adequate protection from severe COVID-19 illness conferred by either vaccine, consistent with public health messaging of other studies.

This study has several limitations. First, potential confounders such as COVID-19 exposure behaviors could not be measured. Second, our definition of immunocompromised status may have missed some of the relatively smaller sub-populations such as those with intrinsic immune deficits or others who do not take immunosuppressive agents or have a history of cancer; our analyses of this group considered timing of 3rd dose (excluding those who obtained their third dose within 2 months) but not size of dose. Third, sub-groups such as nursing home residents were excluded so that the findings could have greater generalizability to community-dwelling individuals living in the U.S. However, the boosted study population comprised predominantly older, white men with high-risk comorbidities, so despite the substantial absolute numbers of participants with female sex, younger age, African American race, Latino ethnicity, and no comorbidities, inferences to these sub-populations (e.g., age, sex, race) should be approached with caution. Fourth, it is unclear whether these findings would apply to 4 or more doses of vaccine, updated booster vaccine products (e.g., bivalent Omicron booster), and other viral variants. Fifth, VHA-distributed mRNA vaccine types were based on facility-specific factors, and it is unknown if the distribution may have affected outcomes. A facility variable was included in the propensity score model to account for potential geographic differences in allocation. Finally, the use of laboratory testing may have decreased over time, which may have led to underestimates of risk within each group but were unlikely to be different between arms.

Comparative mRNA booster effectiveness varied by population and risk group. During a 32-week risk period, a small benefit was found among the high-risk population who received three doses of mRNA-1273 vaccine to prevent hospitalization with COVID-19 pneumonia or death; no benefit was found among the average-risk population and the age >65 sub-group. In an era of vaccination when individuals can choose their booster, it is important to re-evaluate the populations that may benefit from mRNA-specific booster guidelines because of the public health significance and changing times.

## Methods

### Ethics statement

This study complies with all relevant ethical regulations. The institutional review board of the University of California, San Francisco,

**Table 2 | 32-week incidence of and comparative mRNA booster effectiveness against breakthrough COVID-19 and death or hospitalization with COVID-19 pneumonia following vaccination and booster in adults**

| COVID-19 outcome by vaccination and booster | Overall cohort (N = 1,703,189) | | | Average-risk populations (N = 110,485) | | | High-risk populations (N = 1,592,704) | | |
|---|---|---|---|---|---|---|---|---|---|
| | 32-week Risk (95% CI)[a] | Risk diff. (95% CI)[a] | NNV[b] (95% CI)[a] | 32-week Risk (95% CI)[a] | Risk diff. (95% CI)[a] | NNV[b] (95% CI)[a] | 32-week Risk (95% CI)[a] | Risk diff. (95% CI)[a] | NNV[b] (95% CI)[a] |
| | Events per 10,000 persons | Events per 10,000 persons | | Events per 10,000 persons | Events per 10,000 persons | | Events per 10,000 persons | Events per 10,000 persons | |
| **Breakthrough COVID-19 (N = 22,848)** | | | | | | | | | |
| Overall | 149.3 (147.3, 152.1) | - | - | 199.5 (184.1, 215.1) | - | - | 147.1 (143.9, 149.6) | - | - |
| mRNA-1273 ×3 | 137.5 (133.6, 140.9) | Ref. | Ref. | 188.7 (165.7, 217.5) | Ref. | Ref. | 134.7 (130.8, 138.6) | Ref. | Ref. |
| BNT162b2 ×3 | 161.5 (158.1, 165.0) | 23.9 (19.8, 29.9) | 415.5 (334.7, 505.3) | 210.1 (190.4, 229.4) | 21.4 (−12.7, 53.4) | 411.1 (−7,320.8, 4,913.9) | 159.2 (154.9, 162.4) | 24.5[d] (18.7, 29.4) | 414.2 (340.0, 534.7) |
| **Death or hospitalization with COVID-19 pneumonia[c] (N = 1649)** | | | | | | | | | |
| Overall | 10.9 (10.2, 11.8) | - | - | 1.4 (0.6, 2.6) | - | - | 11.5 (10.8, 12.4) | - | - |
| mRNA-1273 ×3 | 9.8 (8.8, 11.1) | Ref. | Ref. | 1.1 (0.1, 2.6) | Ref. | Ref. | 10.3 (9.5, 11.6) | Ref. | Ref. |
| BNT162b2 ×3 | 11.9 (12.9, 10.8) | 2.1 (0.6, 3.3) | 4,936.6 (3,028.8, 16,670.3) | 1.7 (0.4, 3.6) | 0.6 (−1.4, 2.7) | 7,110.7 (−87,369.5, 216,408.7) | 12.5 (11.7, 13.7) | 2.2[d] (0.9, 3.6) | 4,505.9 (2,674.5, 10,665.9) |

[a]CI indicates "confidence interval".
[b]NNV indicates number needed to vaccinate.
[c]Death due to all causes within 30 days of breakthrough COVID-19 infection.
[d]There was evidence of additive interaction for death or hospitalization with COVID-19 pneumonia when comparing the risk differences among high-risk populations with the risk differences among average-risk populations (relative excess risk due to interaction [RERI]: 1.6), but there was no evidence of additive interaction for breakthrough COVID-19 (RERI: 0.0).

**Table 3 | 32-week incidence of and comparative effectiveness against breakthrough COVID-19 and death or hospitalization with COVID-19 pneumonia following vaccination and booster in high-risk sub-groups (non-overlapping)**

| COVID-19 outcome by vaccination and booster | Age ≥ 65 with no high-risk conditions (N = 215,177) | | | High-risk co-morbid conditions[a] (not immunocompromised) (N = 1,188,470) | | | Immunocompromising conditions[b] (N = 189,057) | | |
|---|---|---|---|---|---|---|---|---|---|
| | 32-week Risk(95% CI)[c] | Risk diff.(95% CI)[c] | NNV[d] (95% CI)[c] | 32-week Risk (95% CI)[c] | Risk diff. (95% CI)[c] | NNV[d] (95% CI)[c] | 32-week Risk (95% CI)[c] | Risk diff. (95% CI)[c] | NNV[d] (95% CI)[c] |
| | Events per 10,000 persons | Events per 10,000 persons | | Events per 10,000 persons | Events per 10,000 persons | | Events per 10,000 persons | Events per 10,000 persons | |
| **Breakthrough COVID-19 (N = 21,186)** | | | | | | | | | |
| Overall | 82.4 (76.1, 87.1) | - | - | 136.1 (133.2, 139.0) | - | - | 292.9 (284.2, 303.5) | - | - |
| mRNA-1273 ×3 | 78.2 (71.5, 84.9) | Ref. | Ref. | 123.3 (118.7, 128.1) | Ref. | Ref. | 266.9 (255.0, 281) | Ref. | Ref. |
| BNT162b2 ×3 | 86.5 (79.0, 94.0) | 8.3 (−0.3, 19.4) | 1,162.5 (−9,063.8, 14,110.7) | 148.8 (144.6, 153.7) | 25.5[f] (19.8, 31.4) | 188.5 (145.0, 295.9) | 319.2 (306.1, 331.9) | 52.3[g] (33.8, 69.0) | 188.5 (145.0, 295.9) |
| **Death or hospitalization with pneumonia[e] (N = 1637)** | | | | | | | | | |
| Overall | 3.0 (1.9, 4.5) | - | - | 8.1 (7.2, 9.2) | - | - | 42.6 (39.0, 46.4) | - | - |
| mRNA-1273 ×3 | 3.2 (1.2, 5.8) | Ref. | Ref. | 6.8 (5.6, 8.6) | Ref. | Ref. | 39.2 (34.1, 44.3) | Ref. | Ref. |
| BNT162b2 ×3 | 2.9 (1.7, 4.2) | −0.3, (−3.2, 2.1) | −4,170.3 (−98,032.7, 125,506.1) | 9.4 (10.7, 8.3) | 2.6[f] (0.5, 4.4) | 3,869.6 (2,128.3, 12,063.0) | 45.8 (40.4, 52.1) | 6.6[g](−1.2, 13.5) | 1,411.7 (−7,877.0, 16,932.8) |

[a]As defined by the Center for Disease Control and Prevention, older adults, people with medical conditions who are not immunocompromised, and pregnant and recently pregnant people.
[b]Immunocompromised was defined based on medications and history of cancer (see supplement for list of meds).
[c]CI is an abbreviation for "confidence interval."
[d]NNV indicates number needed to vaccinate.
[e]Death due to all causes within 30 days of breakthrough COVID-19 infection.
[f]There was evidence of additive interaction when comparing the risk differences among those with high-risk co-morbid or immunocompromising conditions with the risk differences among those with high-risk co-morbid conditions with COVID-19 pneumonia: 1.4).
[g]There was evidence of additive interaction when comparing the risk differences among those with immunocompromising conditions with the risk differences of other sub-groups (relative excess risk due to interaction [RERI] for breakthrough COVID-19: 0.2; RERI for death or hospitalization with COVID-19: 0.3; RERI for death or hospitalization with COVID-19 pneumonia: 1.1).

**Table 4 | Cumulative incidence ratios of breakthrough COVID-19 and hospitalization with COVID-19 pneumonia or death comparing Veterans who received mRNA-1273 ×3 with those who received BNT–162b2, overall and by sub-group**

| | Overall cohort (N = 1,703,189) | Average-risk populations (N = 110,485) | High-risk populations (N = 1,592,704) | Age ≥ 65 with no high-risk conditions (N = 215,177) | High-risk co-morbid conditions[a] (not immune-compromised) (N = 1,188,470) | Immuno-compromising conditions[b] (N = 189,047) |
|---|---|---|---|---|---|---|
| COVID-19 outcome by vaccination and booster | Cumulative incidence ratios (95% CI)[c] | Cumulative incidence ratios (95% CI)[c] | Cumulative incidence ratios (95% CI)[c] | Cumulative incidence ratios (95% CI)[c] | Cumulative incidence ratios (95% CI)[c] | Cumulative incidence ratios (95% CI)[c] |
| Breakthrough COVID-19 (N = 21,186) | | | | | | |
| mRNA-1273 ×3 | Ref. | Ref. | Ref. | Ref. | Ref. | Ref. |
| BNT162b2 ×3 | 1.2 (1.2, 1.2) | 1.1 (1.0, 1.3) | 1.2 (1.2, 1.3) | 1.1 (1.0, 1.2) | 1.2 (1.2, 1.3) | 1.2 (1.1, 1.3) |
| Death or hospitalization with COVID-19 pneumonia[d] (N = 1637) | | | | | | |
| mRNA-1273 ×3 | Ref. | Ref. | Ref. | Ref. | Ref. | Ref. |
| BNT162b2 ×3 | 1.3 (1.2, 1.4) | 1.7 (0.5, 5.1) | 1.3 (1.2, 1.4) | 1.6 (0.9, 3.0) | 1.4 (1.2, 1.6) | 1.2 (1.0, 1.4) |

[a]As defined by the Center for Disease Control and Prevention, older adults, people with medical conditions who are not immunocompromised, and pregnant and recently pregnant people.
[b]Immunocompromised was defined based on medications and history of cancer (see supplement for list of meds).
[c]CI indicates "confidence interval".
[d]Death from all causes within 30 days of breakthrough COVID-19 infection.

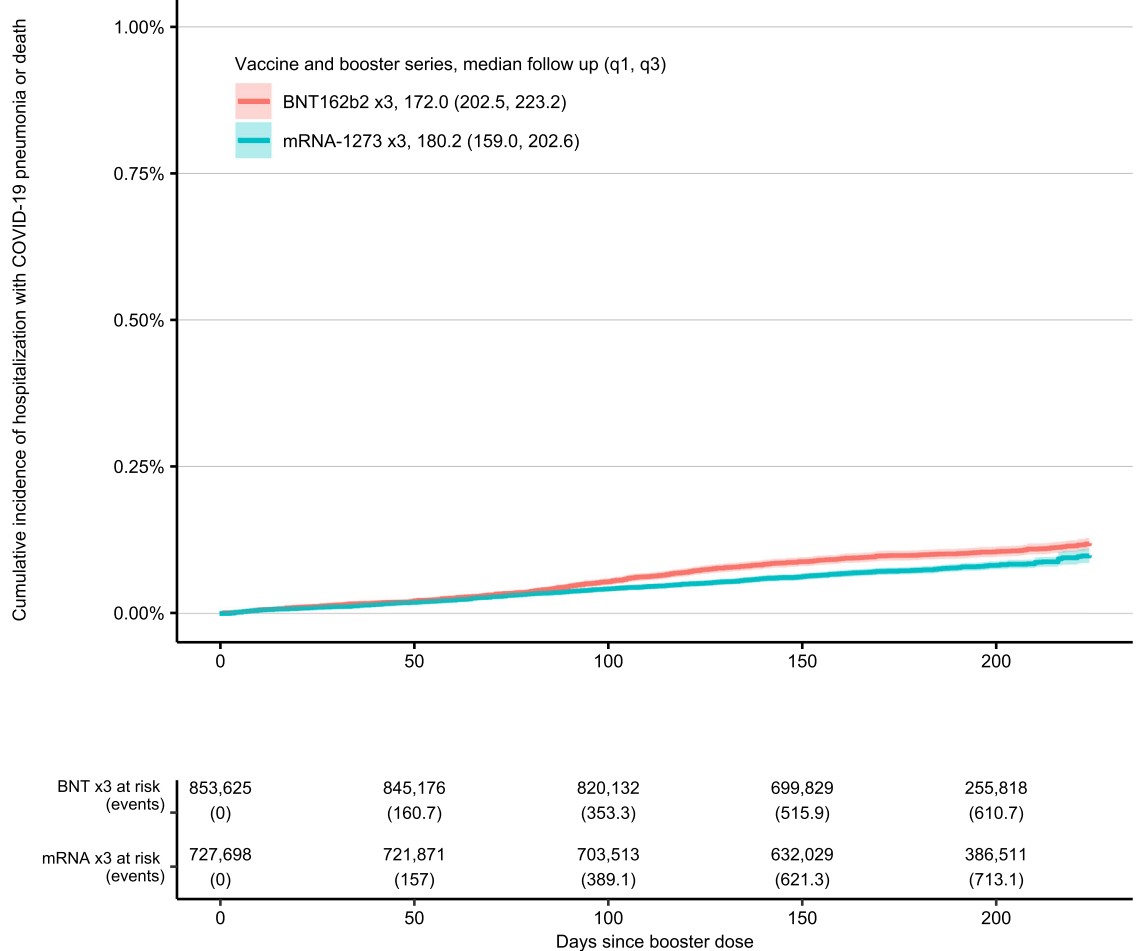

**Fig. 1 | Overall cohort.** 32-week cumulative incidence of death or hospitalization with COVID-19 pneumonia following vaccination and booster with BNT162b2 ×3 and mRNA-1273 ×3 in overall cohort.

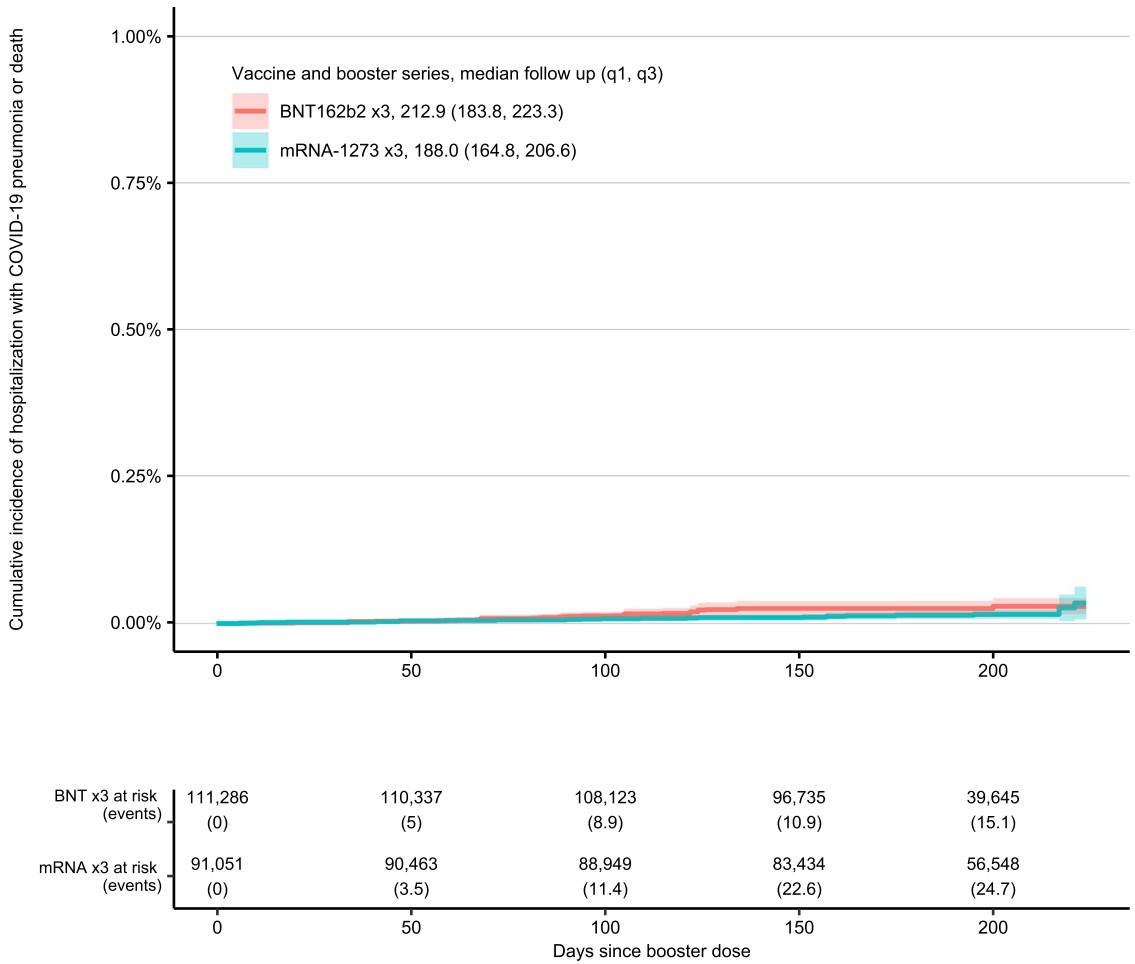

**Fig. 2 | Those with age >65 with no high-risk conditions.** 32-week cumulative incidence of death or hospitalization with COVID-19 pneumonia following vaccination and booster with BNT162b2 ×3 and mRNA-1273 ×3 restricted to those with age >65 with no high-risk conditions.

approved this study and waived requirement for patient consent as it involved no more than minimal risk to participants.

### Study design, setting, and data sources
We used Veterans Health Administration (VHA) Corporate Data Warehouse (CDW)[14] and COVID-19 Shared Data Resource[15] to construct the adult cohort for this target trial emulation. The VHA operates a national health system with co-localized services, including COVID-19 vaccination, laboratory testing, outpatient, and inpatient services. The VHA began administering the initial series of COVID-19 vaccinations after December 11, 2020. The health system provided facilities with the initial series of either mRNA-1273 or BNT162b2 vaccines based on local access to cold chain capacity and geography. The VHA allocated to its facilities either mRNA-1273 or BNT162b2 vaccine for the initial series, so patient preference had no role in the administration of either mRNA vaccine, ultimately resulting in almost all (>90%) Veterans receiving homologous vaccine doses. Booster doses became available on 1 July 2021 and at the time, higher-risk groups, particularly immunocompromised individuals, were among the first to receive a third dose.

### Participants
Adults receiving care at VHA facilities were eligible for inclusion if they had documented receipt of all U.S. Food and Drug Administration (FDA) authorized doses of the initial vaccination series of an mRNA vaccine and subsequently had documented receipt of a third dose between 1 July 2021 and 29 April 2022. The cohort was restricted to participants who had a primary care visit at a VHA facility in 2019 or

2020 to ensure adequate baseline data on health status and capture of clinical outcomes. Those participants living in the community without a record of vaccination who delayed the second dose beyond 6 weeks, and did not receive three doses of either mRNA-1273 or BNT162b2 vaccines were excluded[1]. Participants with non-immunocompromising conditions who obtained their third dose within 5 months and those with immunocompromising conditions who obtained their third dose within 2 months were also excluded[1]. Further inclusion and exclusion criteria are detailed in Supplementary Fig. 1.

The index date (or date of cohort entry) was the date on which the subject received his or her third dose (first booster vaccine). For each person, follow-up started on the day of booster dose and ended on the day of outcome of interest, death, 224 days (32 weeks) after baseline, fourth dose, or the end of the study period (May 30, 2022), whichever happened first. The observation period included predominance of both Delta and Omicron SARS-CoV-2 variants in the US[16].

### Measurements
Exposure was receipt of three doses of mRNA-1273 vaccine or three doses of BNT162b2 vaccine, as extracted from the CDW. The primary outcomes were: (1) breakthrough COVID-19 (symptomatic infection); and (2) the combined endpoint of all-cause death within 30 days after breakthrough infection or hospitalization with a diagnosis of COVID-19 pneumonia.

Breakthrough COVID-19 was defined as a post-booster, laboratory-confirmed, symptomatic COVID-19 diagnosis, using the VA's COVID-19 National Surveillance Tool[17,18] and VA's COVID-19 Shared

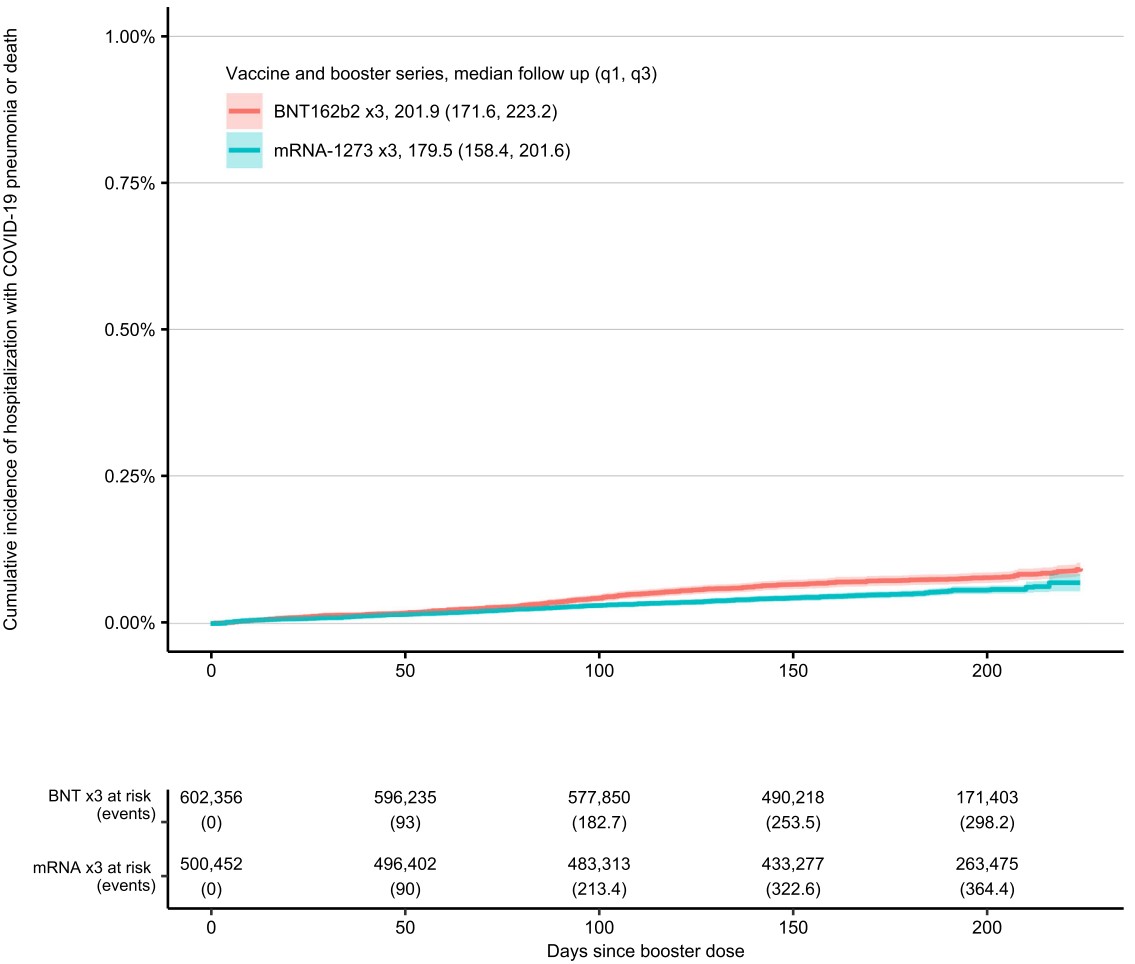

**Fig. 3 | Those with high-risk co-morbid conditions (not immunocompromising conditions).** 32-week cumulative incidence of death or hospitalization with COVID-19 pneumonia following vaccination and booster with BNT162b2 ×3 and mRNA-1273 ×3 restricted to those with high-risk co-morbid conditions (not immunocompromising conditions).

Data Resource[19]. Hospitalization with COVID-19 pneumonia was defined as a diagnosis of COVID-19 pneumonia using ICD-10 code J12.84 in the CDW[20], or documented by the clinical care team in the electronic medical record during hospitalization.

The outcome of COVID-19 pneumonia was verified through a combination of text processing-assisted chart review and ICD-10 codes (see Supplement). Study staff reviewed 10% of charts in duplicate (95% agreement) and flagged unclear diagnoses of pneumonia (e.g., emergency room note was only reference to pneumonia during hospitalization) for adjudication by three clinicians (JDK, SK, DMB). A list of covariates considered to be cohort characteristics or potential confounders were extracted from CDW and are described in the Supplement.

**Statistical analyses**

Baseline covariates of the boosted cohort were compared between participants who received three doses of mRNA-1273 and three doses of BNT162b2. Standardized differences were used to quantify differences in covariates between exposure groups. Among the boosted cohort, the propensity score for receipt of three doses of mRNA-1273 versus BNT162b2 was estimated using logistic regression (see Supplement). Inverse probability of treatment weights (IPTW) were computed as $Z/PS + (1-Z)/(1-PS)$, where $Z$ indicates treatment status ($Z = 1$ for three doses of BNT162b2; $Z = 0$ for three doses of mRNA-1273) and PS indicates the estimated propensity score. The use of these weights allowed us to estimate the average treatment effect (ATE): the effect of moving all subjects from BNT162b to mRNA-1273.

The weighted cumulative incidence function was used to estimate the risk of the outcome from the day of booster dose until end of follow-up and the risk difference (absolute risk reduction). Participants who died prior to developing breakthrough COVID-19 were treated as a competing risk on date of death. Weighted (adjusted) Fine-Gray models were used to estimate cumulative incidence ratios (risk ratios). Robust variance estimators accounted for the within-person homogeneity induced by weighting. The proportionality assumption was tested with a log-log plot and was met. The risk difference was used to estimate the number needed to vaccinate (NNV). The association of booster product with breakthrough COVID-19 and death or hospitalization with COVID-19 pneumonia was examined. Findings were considered statistically significant if the confidence interval did not cross the null value or if the two-sided $p$ value was <0.05.

These analyses were repeated within sub-groups. The overall cohort was divided into average-risk and high-risk sub-groups, with the high-risk population further sub-divided into three non-overlapping sub-groups: (1) age ≥65 years with no high-risk conditions, (2) high-risk co-morbid conditions (excluding immunocompromised or with cancer), and (3) immunocompromising conditions (including cancer). These high-risk sub-groups were pre-specified and based on risk stratification from COVID-19 vaccine guidance set forth by the U.S. Centers for Disease Control and Prevention[1].

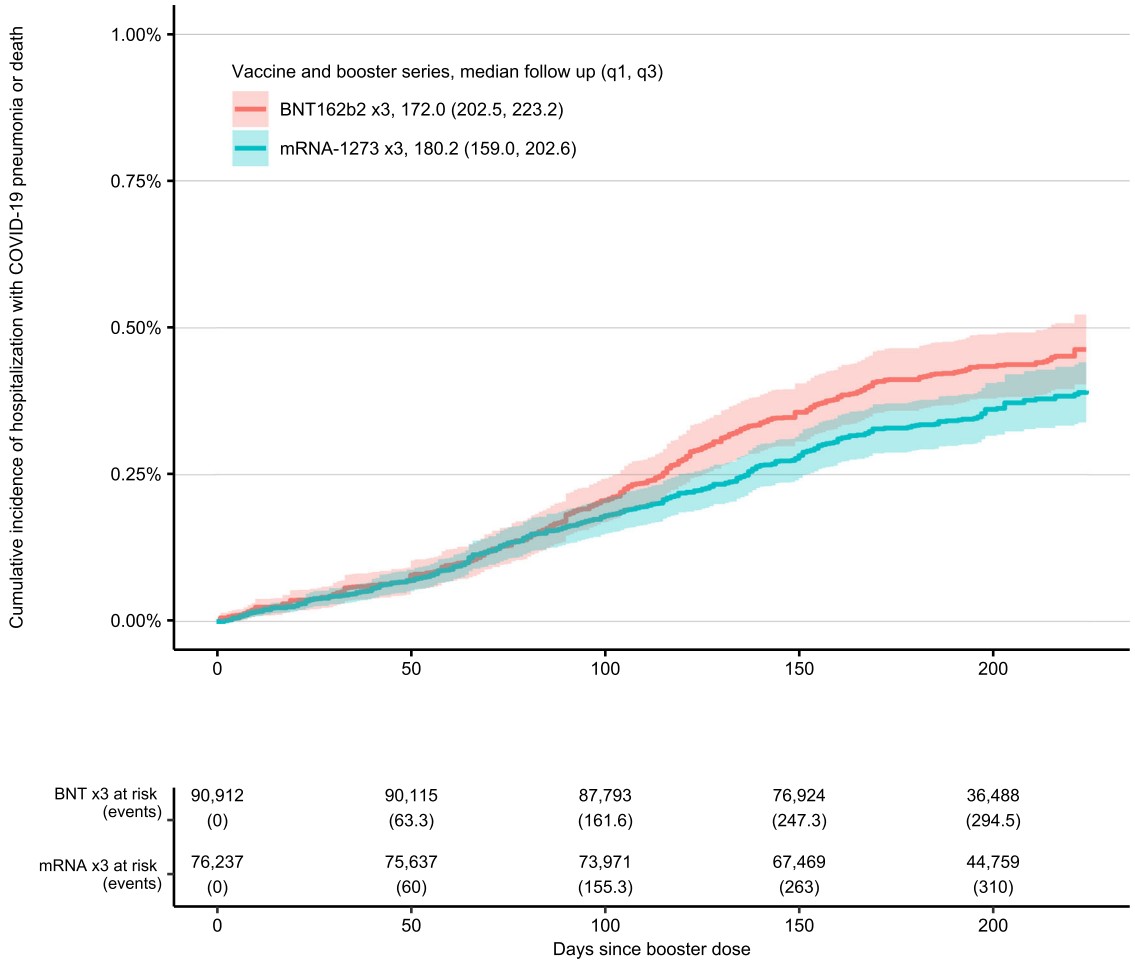

**Fig. 4 | Those with immunocompromising conditions.** 32-week cumulative incidence of death or hospitalization with COVID-19 pneumonia following vaccination and booster with BNT162b2 ×3 and mRNA-1273 ×3 restricted to those with immunocompromising conditions.

Another set of analyses were designed to assess the interaction of booster product by sub-group and by Delta and Omicron predominant variant eras. Interaction was evaluated on the multiplicative and additive scales. The presence of additive interaction was defined as the relative excess risk due to interaction (RERI)[21]. This study evaluated additive interaction because of its relevance to public health and focus on risk differences and NNV. The variant era was based on the predominant variant circulating during a calendar period. The Delta-predominant era was between 1 July 2021 and 30 November 2021. The Omicron-predominant era was between 15 December 2021 and 30 May 2022. An interaction $p$ value cutoff of 0.20 or non-zero RERI determined statistical significance.

In all analyses, standardized differences of each covariate were all <0·01 (Supplementary Table 4). Missing data were handled as a 'missing' category for the variable. Analyses were conducted in RStudio version 1·2·5019, including the survival package.

**Reporting summary**
Further information on research design is available in the Nature Portfolio Reporting Summary linked to this article.

## Data availability

Data to generate the findings of this study are available from the US Department of Veterans Affairs. The raw data are protected and not made freely available due to data privacy laws. More information is available at https://www.virec.research.va.gov. Source data are provided with this paper.

## Code availability

Analytic code is available at github.com/SamuelJLeonard/Kelly-et-all-Nature-Communications[22]. Access to Data: Dr. Kelly and Mr. Leonard had full access to all the data in the study and take responsibility for the integrity of the data and the accuracy of the data analysis.

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

## Acknowledgements

We would like to thank the U.S. Veterans who received vaccine booster doses and contributed data to our study. We appreciate our San Francisco-based chart review team and other VHA employees who have supported aspects of this study. We also thank Amy Markowitz, J.D., for her contributions to the manuscript. This work was supported by the Veteran Affairs Clinical Science Research and Development (IO1 grant number CX002417 to J.D.K. and S.K.) and the National Institute of Allergy and Infectious Diseases (K23 grant number AI146268 to JDK). The sponsor had no role in any of these aspects of the study, including the decision to submit for publication, right to veto publication, or right to control the decision regarding which journal the paper was submitted.

## Author contributions

J.D.K., S.L., W.J.B., K.J.H., A.B., P.C.T., D.M.B., and S.K. conceived and designed the study. J.D.K., S.L., E.N.L., C.C.A., D.M.B., and S.K. contributed to data collection and curation. J.D.K., S.L., E.N.L., C.C.A., P.C.A., D.M.B., and S.K. accessed and verified all data, did the data analysis, drafted the first version of the manuscript, and wrote the final version. J.D.K., S.L., W.J.B., K.J.H., E.N.L., C.C.A., A.B., P.C.T., D.M.B., and S.K. revised the manuscript. All authors approved the final version.

## Competing interests

The authors declare no competing interests.
