## [Peer review file · Nature Communications]

REVIEWER COMMENTS

Reviewer #1 (Remarks to the Author):

This study uses the VA health care system to estimate the comparative effectiveness of homologous 3rd dose of mRNA-1273 versus BNT162b2 conducted during the Delta and Omicron waves. Relative effectiveness of these two booster doses is (or was) an important public health question and of interest. The propensity score matching approach is straightforward and good balance is achieved in this large study.

Comments

Should be Rstudio rather than R.

Were the high risk subgroup analyses pre-specified?

Not sure why the argument to use robust variance estimates is to account for within person homogeneity induced by weighting. I don't know what this means.

Line 285. Not sure what you mean by cumulative incidence increased after 12 weeks post booster. Perhaps cumulative incidence curves diverged after 12 weeks as shown in Figure 1a?

Line 289 'greatly benefit' seems overstated can you reword.

I would mention a limitation of the study is that it's not clear whether the results would apply to 4 or more doses of variant booster vaccines with the current variants.

Reviewer #2 (Remarks to the Author):

In their report, Comparative mRNA booster effectiveness against death or hospitalization with COVID-19 pneumonia across at-risk populations in a national US veteran cohort: a 32-week target trial emulation

study, J.D. Kelly and colleagues describe an interesting study comparing VE of mRNA-1273 versus that of BNT162b2 boosters in preventing hospitalization and death due to COVID-19. Overall, the study is well designed and, while monovalent boosters are no longer recommended in the US, a head-to-head comparison of the VE of mRNA-1273 versus BNT162b2 remains of interest. Furthermore, monovalent boosters continue to be used in global areas with limited access to updated bivalent boosters.

However, I have a few comments:

- 1.) Since a third dose (100 ug mRNA-1273 or 30 ug BNT162b2) is recommended as part of the primary series in IC populations in the US, it is not considered to be a booster in this population. Therefore, the 3rd dose in this population is better described as a 3rd dose rather than as a booster dose. Furthermore, as the 3rd primary dose in this population should be a 100 ug mRNA-1273 dose, it would help to clarify what proportion of participants with IC received 100 ug versus 50 ug mRNA-1273 doses as their 3rd dose in this study and stratify the analysis accordingly (or exclude one subset from analysis if sample size is small).
- 2.) The results would be of greater interest if stratified by variant (at a minimum Delta versus Omicron). The 32-week VE of monovalent COVID-19 mRNA vaccine booster doses is likely much different for Delta than it is for Omicron sub-lineages, particularly more recent sub-lineages, as demonstrated by several prior studies.
- 3.) Immunocompromised variable should include transplant recipients and, ideally, intrinsic immune deficits. Variations on Greenberg JA et al (Annals ATS 2016) algorithm are widely used to evaluate VE among persons with IC: more recent CDC-sponsored studies have updated versions that are very similar (e.g., Izurieta HS et al RZV Real World Effectiveness in the First 2 Years Post Licensure CID 2021).
- 4.) Persons receiving antiretroviral therapy as PreP are unlikely to be IC. HIV infection and/or AIDS diagnosis codes may be more specific (and more comprehensive: not all persons with HIV infection adhere to antiretroviral therapy) for IC.
- 5.) Please clarify if persons with malignancy (solid or hematologic) who are receiving chemotherapy excluded from the non-IC group. Including persons on chemotherapy or other immunosuppressants in the group with comorbidities likely would impact the comparative VE analysis among participants with comorbidities.
- 6.) It would be helpful to add SD pre- and post-IPTW in eTables 1a-f. Most pre-IPTW variables appear to be well-balanced. Nonetheless, it is helpful to compare pre- vs post-IPTW differences.
- 7.) The study population is older and has more comorbidities than many populations that may limit the generalizability to other settings.

I hope that these comments are helpful!

Reviewer #1 (Remarks to the Author):

This study uses the VA health care system to estimate the comparative effectiveness of homologous 3rd dose of mRNA-1273 versus BNT162b2 conducted during the Delta and Omicron waves. Relative effectiveness of these two booster doses is (or was) an important public health question and of interest. The propensity score matching approach is straightforward and good balance is achieved in this large study.

Comments

Should be Rstudio rather than R.

Response → Thank you. We correct this to RStudio on Line 208 of the manuscript revision.

Were the high-risk subgroup analyses pre-specified?

Response → Excellent question. Yes, the high-risk subgroup analyses were pre-specified. We now clearly state this in the third paragraph of Statistical Analysis section of the Methods along with a description of the sub-groups. The revised paragraph reads as follows: "These analyses were repeated within sub-groups. The overall cohort was divided into average-risk and high-risk sub-groups, with the high-risk population further sub-divided into 3 non-overlapping sub-groups: 1) age ≥ 65 years with no high-risk conditions, 2) high-risk co-morbid conditions (not including immunocompromised or with cancer), and 3) immunocompromising conditions (including cancer). These high-risk sub-groups were pre-specified and based on risk stratification from COVID-19 vaccine guidance set forth by the U.S. Centers for Disease Control and Prevention and VHA."

Not sure why the argument to use robust variance estimates is to account for within person homogeneity induced by weighting. I don't know what this means.

Response → This is good question and is part of the statistical approach to estimate propensity scores. When we computed inverse probability of treatment weights, we estimated a pseudo-population, meaning that each individual had two observations (one with three doses of BNT162b2 and the other with three doses of mRNA-1273). The pair of observations per individual can be correlated, so robust variance estimators are commonly used to statistically account for this correlation (within-person homogeneity).

Line 285. Not sure what you mean by cumulative incidence increased after 12 weeks post booster. Perhaps cumulative incidence curves diverged after 12 weeks as shown in Figure 1a?

Response → This suggested phrasing is helpful, so we've revised the sentence to read as follows: "In the overall cohort, the cumulative incidence of death or hospitalization due to COVID-19 pneumonia by booster product was similar until week 12 at which time the cumulative incidence curves diverged after 12 weeks as shown in **Figure 1a**."

Line 289 'greatly benefit' seems overstated can you reword.

Response → We agree with the reviewer and have tempered our language. Further, in the revised manuscript with the updated population and cohort selection criteria, we found that the absolute risks of the outcomes (breakthrough COVID-19, death or hospitalization from COVID-19 pneumonia) converged more towards the null for the overall cohort and high-risk sub-groups. As a result, our conclusion now reads as follows: “During a 32-week risk period, a marginal benefit from receiving three doses of mRNA-1273 vaccine to prevent hospitalization with COVID-19 pneumonia or death was conferred on sub-groups with high-risk comorbid or immunocompromising conditions; no benefit was found among average-risk and age ≥ 65 y sub-groups.” This message is reflected throughout the manuscript, including places where “benefit” is mentioned.

I would mention a limitation of the study is that it's not clear whether the results would apply to 4 or more doses of variant booster vaccines with the current variants.

Response → We appreciate this point and have revised the limitations section to include the suggestion with the following sentence: “It is unclear whether these findings would apply to 4 or more doses of vaccine, updated booster vaccine products (e.g., bivalent Omicron booster), and other viral variants.”

Reviewer #2 (Remarks to the Author):

In their report, Comparative mRNA booster effectiveness against death or hospitalization with COVID-19 pneumonia across at-risk populations in a national US veteran cohort: a 32-week target trial emulation study, J.D. Kelly and colleagues describe an interesting study comparing VE of mRNA-1273 versus that of BNT162b2 boosters in preventing hospitalization and death due to COVID-19. Overall, the study is well designed and, while monovalent boosters are no longer recommended in the US, a head-to-head comparison of the VE of mRNA-1273 versus BNT162b2 remains of interest. Furthermore, monovalent boosters continue to be used in global areas with limited access to updated bivalent boosters.

However, I have a few comments:

1.) Since a third dose (100 ug mRNA-1273 or 30 ug BNT162b2) is recommended as part of the primary series in IC populations in the US, it is not considered to be a booster in this population. Therefore, the 3rd dose in this population is better described as a 3rd dose rather than as a booster dose. Furthermore, as the 3rd primary dose in this population should be a 100 ug mRNA-1273 dose, it would help to clarify what proportion of participants with IC received 100 ug versus 50 ug mRNA-1273 doses as their 3rd dose in this study and stratify the analysis accordingly (or exclude one subset from analysis if sample size is small).

Response → Thank you for your insight about the various doses in immunocompromised populations in the US. While the VA dataset has many strengths, the size of the vaccine dose is not an available variable, so we have included this point in our limitations section. However, we also decided to conduct a limited investigation to clarify the proportion receiving a smaller versus larger dose. We drew a random sample of 25 participants with immunocompromising conditions (IC) who received mRNA-1273 vaccine dose 3 and another 25 participants with IC who received dose 4. We then conducted a chart review to understand the proportion who

received the 3rd primary dose versus a booster dose. See Table below. For dose 3, we found that the majority in IC population received their booster dose rather than 3rd primary dose; and for dose 4, almost all received the expected size of the booster dose. Since the 3rd dose encompasses use of both dose sizes, we have replaced the word “booster” to “3rd dose” throughout the manuscript and referenced this particular limitation in the IC population in the limitations section, reading as follows: “Our definition of immunocompromised may have missed some of the relatively smaller sub-populations such as those with intrinsic immune deficits or others who do not take immunosuppressive agents or have a history of cancer; our analyses of this group considered timing of 3rd dose (excluding those who obtained their third dose within 2 months) but not size of dose.”

Table. Chart review of 3rd and 4th mRNA-1273 vaccine dose received by participants with immunocompromising conditions. We drew a random sample of 25 participants for each mRNA-1273 vaccine dose.

	Dose 3 (N=25)	Dose 4 (N=25)
50 ug mRNA-1273 dose, n (%)	10 (40)	22 (88)
100 ug mRNA-1273 dose, n (%)	8 (32)	1 (4)
Unclear, n (%)*	7 (28)	2 (8)

*Reasons for Unclear: Non-VA (3), Location “Other” – unable to determine location (3), No record (2), Unclear (ml without dose reported) (1)

2.) The results would be of greater interest if stratified by variant (at a minimum Delta versus Omicron). The 32-week VE of monovalent COVID-19 mRNA vaccine booster doses is likely much different for Delta than it is for Omicron sub-lineages, particularly more recent sub-lineages, as demonstrated by several prior studies.

Response → We agree that stratification by variant is an important consideration, so we have performed this analysis. On Lines 237 to 239 in the revision manuscript, we stated, “Comparative effects of the third mRNA vaccine dose (BNT162b2 versus mRNA-1273) were sustained over eras of Delta and Omicron predominant variants (**eTable 4**.)” These results can be found in the Supplement. Given that there was no evidence of interaction by variant, we retained presentation of the results, regardless of variant, for simplicity and clarity.

3.) Immunocompromised variable should include transplant recipients and, ideally, intrinsic immune deficits. Variations on Greenberg JA et al (Annals ATS 2016) algorithm are widely used to evaluate VE among persons with IC: more recent CDC-sponsored studies have updated versions that are very similar (e.g., Izurieta HS et al RZV Real World Effectiveness in the First 2 Years Post Licensure CID 2021).

Response → We appreciate this point, so we noted it in the limitations section by writing the following sentence: “Our definition of immunocompromised status may have missed some of the relatively smaller sub-populations such as those with intrinsic immune deficits or others who do not take immunosuppressive agents or have a history of cancer.”

4.) Persons receiving antiretroviral therapy as PrEP are unlikely to be IC. HIV infection and/or AIDS diagnosis codes may be more specific (and more comprehensive: not all persons with HIV infection adhere to antiretroviral therapy) for IC.

Response -> This is an excellent point, and the reviewer was correct. A proportion of individuals were receiving antiretroviral therapy and did not have a diagnosis code for HIV infection and/or AIDS, suggesting that they were receiving antiretroviral therapy as pre-exposure prophylaxis (PrEP). Since these individuals were unlikely to be immunocompromised, we moved these individuals from our sub-group of immunocompromising conditions to their appropriate lower risk sub-group. We also performed a sensitivity analysis with and without these individuals included in the sub-group and found no differences in results.

5.) Please clarify if persons with malignancy (solid or hematologic) who are receiving chemotherapy excluded from the non-IC group. Including persons on chemotherapy or other immunosuppressants in the group with comorbidities likely would impact the comparative VE analysis among participants with comorbidities.

Response -> We appreciate the opportunity to clarify. The high-risk subgroups were non-overlapping, so persons with cancer who are receiving chemotherapy were excluded from the non-immunocompromised group. We state details of the non-overlapping nature of the high-risk sub-groups in the Statistical Analysis section of the Methods and have attempted to make this more explicit in the revised version of the following sentence: "The overall cohort was divided into average-risk and high-risk sub-groups, with the high-risk population further sub-divided into 3 non-overlapping sub-groups: 1) age \geq 65 years with no high-risk conditions, 2) high-risk comorbid conditions (not including immunocompromised or with cancer), and 3) immunocompromising conditions (including cancer)." Further, we have added the word "non-overlapping" as qualifier of "high-risk sub-groups" in various places throughout the manuscript, including sub-section headers and table titles to provide additional clarity.

6.) It would be helpful to add SD pre- and post-IPTW in eTables 1a-f. Most pre-IPTW variables appear to be well-balanced. Nonetheless, it is helpful to compare pre- vs post-IPTW differences.

Commented [KD1]: Remind Sam about pre-IPTW variables

Response -> We have included SD of pre- and post-IPTW variables in eTables 1a-f. These variables remain well-balanced.

7.) The study population is older and has more comorbidities than many populations that may limit the generalizability to other settings.

Response -> We agree with this point and have explicitly highlighted it in the following revised sentence of the limitations section: "The boosted study population comprised predominantly older, white men with high-risk comorbidities, so despite the substantial absolute numbers of participants with female sex, younger age, African American race, Latino ethnicity, and no comorbidities, inferences to these subpopulations (e.g., age, sex, race) should be approached with caution."

I hope that these comments are helpful!

Response → Yes, they have been helpful—thank you!

REVIEWERS' COMMENTS

Reviewer #2 (Remarks to the Author):

The authors' response have addressed my comments adequately. thank you for your thoughtful responses and edits when indicated. As noted in my initial review, I feel that these results of this well-designed study are of interest to readers.